# Effect of a Short Course on Improving the Cadres’ Knowledge in the Context of Reducing Stunting through Home Visits in Yogyakarta, Indonesia

**DOI:** 10.3390/ijerph19169843

**Published:** 2022-08-10

**Authors:** Tri Siswati, Slamet Iskandar, Nova Pramestuti, Jarohman Raharjo, Muhammad Primiaji Rialihanto, Agus Kharmayana Rubaya, Bayu Satria Wiratama

**Affiliations:** 1Department of Nutrition, Politeknik Kesehatan Kemenkes Yogyakarta, Tata Bumi No. 3, Banyuraden, Gamping, Sleman, Yogyakarta 55293, Indonesia; 2Pusat Unggulan Iptek Inovasi Teknologi Terapan Kesehatan Masyarakat, Politeknik Kesehatan Kemenkes Yogyakarta, Tata Bumi No. 3 Banyuraden, Gamping, Sleman, Yogyakarta 55293, Indonesia; 3Balai Litbang Kesehatan Banjarnegara, Selamanik No. 16 A, Banjarnegara 53415, Indonesia; 4Department of Environmental Health, Politeknik Kesehatan Kemenkes Yogyakarta, Tata Bumi No. 3, Banyuraden, Gamping, Sleman, Yogyakarta 55293, Indonesia; 5Department of Epidemiology, Biostatistics and Population Health, Faculty of Medicine, Public Health and Nursing, Universitas Gadjah Mada, Yogyakarta 55281, Indonesia; 6Graduate Institute of Injury Prevention and Control, College of Public Health, Taipei Medical University, Taipei 11031, Taiwan

**Keywords:** cadres, stunting, children’s growth, development, monitoring, IYCF, home visits

## Abstract

**Background:** Stunting is primarily a public health concern in Low- and Middle-Income Countries (LMIC). The involvement of Integrated Health Service Post (Indonesian: Posyandu) cadres is among the strategies to combat stunting in Indonesia. **Objective:** This study aimed to determine the effect of a short course on cadres’ knowledge. **Method:** A single group pre-test post-test design was conducted in Yogyakarta, Indonesia, from March to May 2022. Thirty cadres were selected based on the following criteria: willingness to participate, the number of stunted children in their Posyandu, able to read and write, and full attendance at the short course. The knowledge scores were measured by a questionnaire using true and false answers after a short course (post-test 1) and 4 weeks later (post-test 2). We apply STATA 16 to calculate the Mean Difference (MD) using a t-test and a Generalized Estimated Equation (GEE). Furthermore, the adequacy of the short course was evaluated with in-depth interviews. **Result:** GEE analysis showed that after controlling for age, education, occupation, and years of experience, the short course improved cadres’ knowledge significantly on post-tests 1 and 2, i.e., knowledge regarding Children Growth Monitoring (CGM) (Beta = 6.07, 95%CI: 5.10–7.03 and Beta = 8.57, 95%CI: 7.60–9.53, respectively), Children Development Monitoring (CDM) (Beta = 6.70, 95%CI: 5.75–7.65 and Beta = 9.27, 95%CI: 8.31–10.22, respectively), and Infant Young Children Feeding (IYCF) (Beta = 5.83, 95%CI: 4.44–7.23 and Beta = 11.7, 95%CI: 10.31–13.09, respectively). Furthermore, the short course increased their self-efficacy, confidence, and ability to assist stunted children through home visits. **Conclusion:** The short courses consistently and significantly boosted cadres’ knowledge of CGM, CDM, and IYCF, and appropriately facilitated cadres in visits to the homes of stunted children’s home.

## 1. Introduction

Stunting is a chronic malnutrition problem faced by developing countries [1]. Indonesia has a targeted 14% reduction in impaired growth and development [2], following the goal of the World Health Assembly, which is set at 40% by 2024 [3]. Furthermore, after Bali and Jakarta at 17.3% and 10.9%, respectively, Yogyakarta is the province with the third lowest prevalence of stunting in Indonesia, at 17.3% [4]. Although this percentage is included in the mild category (<20%) [5], the disparity of stunted children in Yogyakarta is vast, with a range from 4.6% (Depok Sub-district, Sleman District) to 24.4% (Dlingo Sub-district, Bantul District).

Cadres are local community health volunteers selected by the residents based on their ability, integrity, loyalty, and commitment to improving community health status [6], and play a role in the development of strategies to counteract stunting [7]. Cadres are usually trained to identify individual and community health problems; hence, they can engage in health promotion, provide counseling, and refer medical problems to health care facilities [6]. Cadres continually undergo training to maintain and improve their knowledge and skills in providing services in the community [8]. Previous research has shown that training increases cadres’ responsibility for self-medication [9], improves health service delivery [10], and increase cadres’ capacity to deal with mental disorder patients [11]. 

This study offered a short course to cadres as a debriefing before they assisted with stunted children during home visits. The assistance rendered to families with children at risk of stunted growth is comprehensive, and comprises detailed measures aiming to overcome malnutrition and other health problems. Furthermore, several studies have shown that assistance is effective in increasing dietary diversity scores and identifying malnutrition [12], improving healthcare access [13], increasing breastfeeding success [14], reducing early complementary feeding for children [14], promoting healthy practices [15], increasing body weight, and improving children development [16]. The Indonesian Presidential Regulation Number 72 of 2021 addresses the subject of aiding families at risk of stunting with the aim of improving access to information and services through counseling, referral services facilitation, and social assistance programs [17].

In this research, we used the Integrated Community Case Management (iCCM) theory, which focuses on the most cost-effective and evidence-based child survival strategies, with the aim of saving the lives of infants and children and delivering curative health treatment to children in inaccessible regions [18]. The WHO has urged countries to adopt and promote policies and programs with strong community-based components to deliver interventions for diarrhea, malaria, pneumonia, newborn care, and severe malnutrition, while also improving services at primary health care facilities [19]. Previous studies proved that iCCM succes to improve the health of children, as reported in Sub-Saharan Africa, which experienced a 63% reduction in the annual mortality of children caused by malaria, pneumonia, and diarrhea [20], other countries in RAcE project areas [18], and child infection and malnutrition in Kenya [6,21]. This study reconstructed the iCCM framework based on the local administration of pneumonia home case management [21], as detail described in Figure 1.

This study aimed to determine the effect of a brief course on health cadres’ knowledge of Children Growth Monitoring (CGM), Children Development Monitoring (CDM), and Infant Young Children Feeding (IYCF) to improve health service delivery for stunted children through home visits in Yogyakarta.

## 2. Materials and Methods

### 2.1. Study Design

This study was conducted using a single group intervention pre-test post-test design in the area of highest stunting prevalence in DIY, including two villages (Muntuk and Jatimulyo), Dlingo Sub-district, Bantul Regency, Yogyakarta, Indonesia, from March to May 2022. 

### 2.2. Study Procedures 

As an intervention, participants attended a two-day health education course that combined theory and simulation addressing CDM, CGM, and IYCF. The theory was taught to big groups or classes, whereas the simulation was taught to six-person subgroups. The knowledge and practice were measured at baseline and two follow-ups (after training and 4 weeks later). During this study, cadres made 30–60 min home visits to educate mothers about how to monitor growth, interpret the growth curve, diagnose growth failure, provide developmental stimulation, assess children’s development, and practice IYCF based on the age of the child. Through face-to-face meetings, cadres were reminded of their tasks, the appropriateness of their to-do list, and any difficulties they discovered during home visits. The supervisors consisted of dietitians, midwives, and researchers. Using in-depth interviews, the adequacy of this short course in implementing home visits was evaluated. The study flowchart is depicted in Figure 2.

The short course evaluation was conducted by in-depth interviews with five key informants from each village. The aim was to assess the adequacy of the short course to improve health service delivery for stunted children through home visits. 

### 2.3. Participants

We recruited health cadres as participants, determined purposively using with the following criteria: health cadres in the local sub-village area with a high number of stunted children in Posyandu, Dlingo Sub-district, able to read and write, and able to attend the full course. As a result, 30 cadres were selected as participants from 2 villages, Muntuk and Jatimulyo. The evaluation was conducted by in-depth interviews with 5 informants from each village. The aim was to assess the adequacy of the short course to improve health service delivery for stunted children through home visits. Cadres were able to withdraw at any stage of this study. 

### 2.4. Data Collection 

The knowledge of CGM, CDM, and IYCF were collected with 30 questions using true or false answers. The answers were assigned a score of 1 when correct and 0 for incorrect, then weighted, resulting in a fully correct score of 100. Pearson correlation and Cronbach’s Alpha were used to assess construct and concurrent validity, respectively. The correlation coefficient was r = 0.36, and reliability was 0.99 using Cronbach’s Alpha. An example of the web-based questionnaire is shown in Figure 3. 

The adequacy of the short course in implementing home visit assistance for stunted children was evaluated by in-depth interviews using the Theoretical Framework of Acceptability (TFA). 

### 2.5. Data Management and Analysis

The quantitative data were analyzed with a *t*-test to determine the difference in cadres’ knowledge before and after the short course. Furthermore, the GEE test was also conducted to analyze repeated data. Qualitative data were analyzed through content analysis based on seven themes, namely, affective attitude, burden, ethics, perception of effectiveness, intervention coherence, opportunity cost, and self-efficacy.

## 3. Results

### 3.1. Baseline Characteristics

Most of the cadres had full-time senior high school education (63.3%); were housewives (60%) and individuals with more than 10 years of experience (60%), trained in IYCF (100%), and trained in growth monitoring (100%); and had certified competence. Table 1 shows the details of these data.

### 3.2. Short Course

The offline short course was delivered using classical theory and simulation with sub-groups. Delivery of knowledge about CGM, CDM, and IYCF was undertaken by local experts. The intervention was conducted in a meeting room with ample space and a calm atmosphere in the middle of the forest. Furthermore, the adequate infrastructure included projectors and attractive slides, learning support such as MCH books, anthropometric tools, checklists for children’s development, and food ingredients. The short course was followed with enthusiasm and high motivation because offline coaching activities were suspended for some time due to the COVID-19 pandemic.

The *t*-test shows that the short course consistently increased cadres’ knowledge regarding CGM, CDM, and IYCF in post-tests 1 and 2. Table 2 shows detail of the results.

In general, short course intervention was able to increase participants’ knowledge regarding all outcome variables. The increase was consistent through four weeks after the short course ended. The analysis shows that a short course increased cadres’ knowledge regarding CGM (Beta = 8.57, 95%CI: 7.58–9.56), CDM (Beta = 9.27, 95%CI: 8.28–10.25), and IYCF (Beta = 11.7, 95%CI: 10.55–12.85). The increase was also statistically significant after controlling for age, education, occupation, and years of experience. Table 3 shows details of the results.

After completing the short course, participants conducted a home visit for stunted children’s families at a frequency of once per week for 4 weeks. Supervisors provided offline meetings with cadres every two weeks to maintain implementation of home visit requirements, explore constraints, and propose solutions. In the first supervision, cadres reported that there was a slight refusal by mothers regarding the nutritional status of their stunted children. However, they later received the home visit program after being provided with an explanation. Finally, cadres reported that maternal impressions were positive and that mothers would follow the cadres’ recommendation for children’s health. The home visit was successful overall. Although the home visit was about to expire, the maternal children demanded a longer period of assistance.

Next, we conducted in-depth interviews to assess the effectiveness of the short course after conducting a home visit. Results are detailed in Table 4.

Cadres were asked about the adequacy of the short course for the implementation of the home visits on affective attitude, and they stated that:

“We are pleased with this short course because the duration is sufficient, not too long, and allows us to schedule a home visit.”(SR, 31 years old)

Several participants stated that, in terms of the burden, cost, and effectiveness of the program, they felt there was an additional task in visiting children. However, it was believed that home visit assistance for children was more effective than communal mother education. As stated:

“At first, we had to fix an appointment with the mother, but now it is more accessible through a cellphone. However, we are happy because home visits make us understand the condition of children and their families, hence, it won’t be intensive when we meet at the Integrated Health Service Post.”(TS, 45 years old)

The short course also increased the cadres’ confidence as the knowledge and skills were transferred to mothers:

“I’m becoming more courageous in assisting children. During the home visit, I recalled how we taught monitoring of children’s growth and development, IYCF, and the information was transferred to the mothers until she was understood.”(D, 42 years old)

These opinions imply that the short course has a sufficiently positive impact on cadres to assist stunted children through home visits.

## 4. Discussion

Cadres are essential in bridging health workers with the community. These cadres allow the community to obtain information on health, prevention of diseases and nutritional problems, CGM, CDM, and appropriate IYCF for prevention of stunting [22]. In Indonesia, community empowerment is conducted following the Ministry of Rural Affairs Regulation concerning Priority Use of Village Funds Number 19 of 2017 point 9, by involving health cadres in public health promotion and healthy living [23], via the implementation of the 3rd pillar, namely, the convergence, coordination, and consolidation of national programs [24]. 

The cadres’ performance was closely related to their characteristics. In this study, most were adults, married, had high school education, were housewives, had over 10 years of experience, had previously received a variety of training, and all participated in a complete short course. These findings are consistent with prior research that suggests factors supporting cadres’ performance include age, marital status, knowledge, skills, education, role as housewives related to free time in community health promotion programs [7], and working duration [24,25,26]. Additionally, Bantul- Indonesia has implemented a competency test for cadres, including theory, practice, and counseling exam [27]; all cadres in the current study were certified.

Results show that the short course significantly increased the cadres’ knowledge in post-tests 1 and 2. However, the increase was greater in post-test 2. Past studies in Sulawesi Indonesia [28] and a multi-country study in Ghana, Malawi, Nigeria, Kenya, Tanzania, and Sierra Leone [29] has showed similar results. There are several reasons why a short course may increase knowledge among participants. Firstly, the results show that the cadres implement the knowledge gained in the short course through repeated home visits for stunted children. Second, they have the opportunity to improve their knowledge [30] by teaching mothers how to appropriately monitor the children’s growth and development, in addition to IYCF. Third, knowledge accompanied by practicing has a 90% impact on learning outcomes [31]. Another reason for the intervention’s success is the effectiveness and satisfaction of face-to-face learning [32], which allows participants and informants to interact socially and provide mutual support [33], in combination with their motivation and enthusiasm for participating [34], a supportive environment, and appropriate infrastructure [33,34]. After controlling for age, education, occupation, and years of experience, multivariate analysis reveals that the short course has a significant effect. Previous research showed that most cadres had good knowledge about the early detection of malnutrition in children [35,36]. Furthermore, the level of education also contributes to the increase in their capacity, knowledge, ability, and skills in monitoring growth and development [7,15].

In this study, the short course positively impacted the effectiveness of home visits based on various perspectives including emotions, feelings, values, appreciation, and motivation. The cadres enthusiastically help children through home visits because mothers are provided with solutions that are specific to their health problems. Mothers’ positive responses to home visit interventions motivate cadres, and the mutually reinforcing reciprocal relationship between cadres and mothers aids in the effective achievement of a reduction in stunting [6]. Obviously, stunting prevention measures require financial resources, which can be supplied from the village fund [36].

As a whole, the short course was found to have a positive impact on assisting stunted children through home visits in terms of the self-efficacy, affective attitude, perception of effectiveness, and self-confidence of cadres in teaching mothers about children’s health. Furthermore, the understanding and practice of mothers in providing healthcare to their children, and good practice of CGM, CDM, and IYCF, will improve their health status and help to overcome stunting problems.

## 5. Limitation 

This study has limitations due to the lack of a control group, and because several other variables related to cadre knowledge were not examined, such as motivation, remuneration, support system, environment, and concern of local government. This study focused on an area having a high prevalence of stunting, so different results may be found if the study is conducted in areas having a lower prevalence of stunting.

## 6. Conclusions

The short course was found to significantly improve the cadres’ understanding of CGM, CDM, and IYCF, and to enhance their effective attitude, perception of effectiveness, and self-efficacy. In light of these findings, any training provided to health volunteers must be supported by direct implementation in the target population in order to boost the positive impact and enhance the effectiveness of the health program. As a sign of the village government’s commitment to addressing stunting, this intervention can be sustained by allocating a budget for stunting from village funds.

## Figures and Tables

**Figure 1 ijerph-19-09843-f001:**
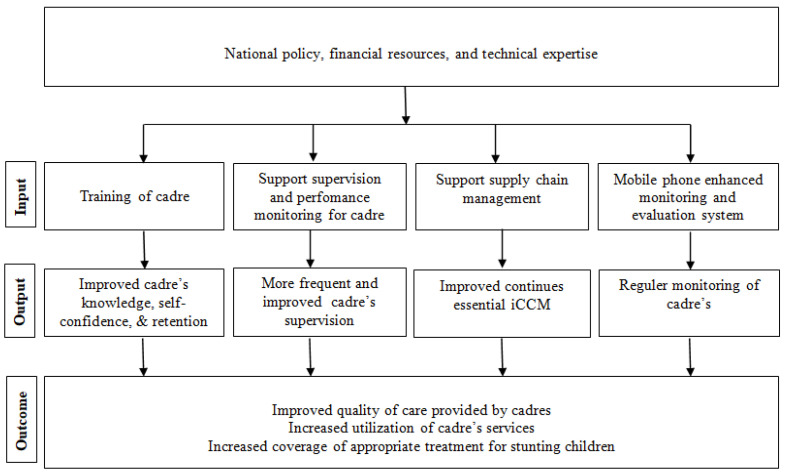
iCCM implementation process and conceptual framework.

**Figure 2 ijerph-19-09843-f002:**
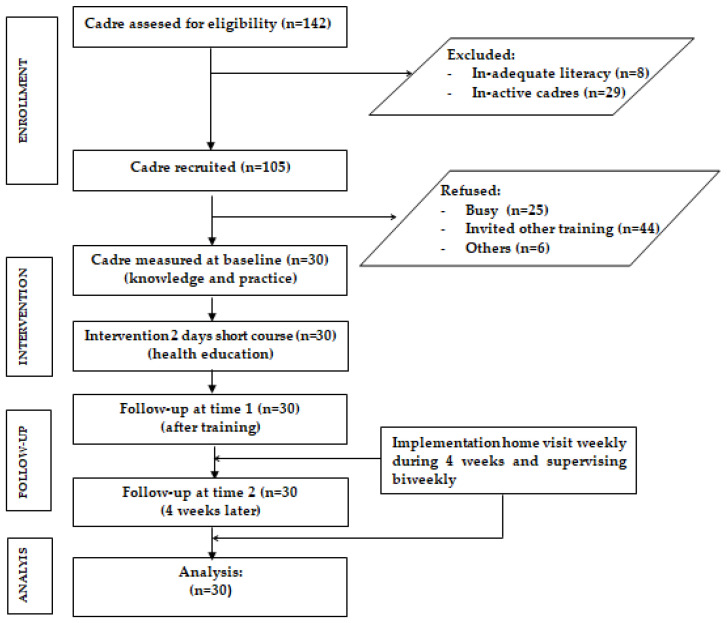
Flow diagram of intervention stage.

**Figure 3 ijerph-19-09843-f003:**
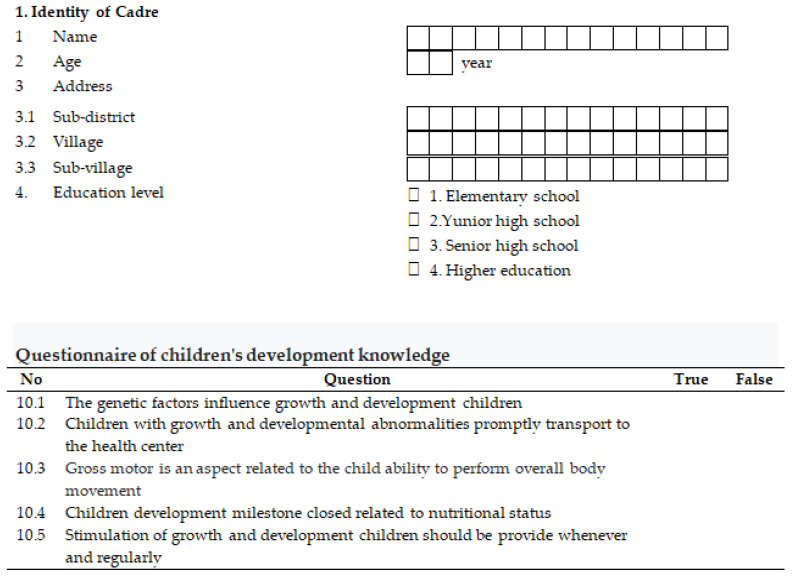
Screenshot of the web-based questionnaire.

**Table 1 ijerph-19-09843-t001:** Characteristics of cadres.

Variables	n	%
Age (years old)		
<30	6	7.0
30–40	12	40.0
>40	18	53.0
Marital Status		
Married	30	100.0
Formal Education		
Junior High School	9	30.0
Senior High School	19	63.3
University	2	6.7
Occupation		
Farmer	8	26.7
Self-employed	4	13.3
Housewife	18	60.0
Years of role as cadres (years)		
<5	7	23.3
6–10	5	16.7
>10	18	60.0
History of training		
IYCF	30	100.0
Growth monitoring	30	100.0
Cadre competency certification	30	100.0
Take a short course completely	30	100.0

**Table 2 ijerph-19-09843-t002:** The impact of short course on cadre knowledge.

Variables	CGM	CDM	IYCF	Average
Pretest	71.50 ± 1.41	70.87 ± 1.96	71.33 ± 1.32	71.23 ± 0.75
Post-test 1	77.57 ± 2.34	77.57 ± 1.96	77.17 ± 2.81	77.43 ± 1.29
Post-test 2	80.07 ± 2.02	80.13 ± 2.16	83.03 ± 3.51	81.08 ± 1.72
Post-test 1—Pretest Effect size 95%CI	6.07 * (5.09–7.04)	6.70 * (5.67–7.73)	5.83 * (4.44–7.22)	6.20 * (5.68–6.72)
Post-test 2—Pretest Effect size 95% CI	8.57 * (7.58–9.56)	9.27 * (8.28–10.25)	11.7 * (10.55–12.85)	9.84 * (9.16–10.53)

* *p*-value <0.05. CGM: Children Growth Monitoring. CDM: Children Development Monitoring. IYCF: Infant Young Children Feeding.

**Table 3 ijerph-19-09843-t003:** Multivariate analysis of short course impact on cadres’ knowledge using GEE.

Variable ^2^	CGM ^1^	CDM ^1^	IYCF ^1^	Average ^1^
Post-test 2	8.57 * (7.60–9.53)	9.27 * (8.31–10.22)	11.7 * (10.31–13.09)	9.84 * (9.17–10.52)
Post-test 1	6.07 * (5.10–7.03)	6.70 * (5.75–7.65)	5.83 * (4.44–7.23)	6.20 * (5.52–6.88)
Pretest	Ref	Ref	Ref	Ref
Cons	71.71	73.47	69.41	71.53
QIC	373.938	428.937	601.954	179.888

^1^ adjusted β coefficient (95%CI). ^2^ controlled variables of age, education, occupation, and years of role as cadre. * *p*-value <0.05. CGM: Children Growth Monitoring. CDM: Children Development Monitoring. IYCF: Infant Young Children Feeding.

**Table 4 ijerph-19-09843-t004:** Adequacy of a short course for cadres in implementing home visits.

Aspect	Opinion
Affective attitude	Cadres felt excited and were more intensive in assisting stunted children. They can educate mothers more comprehensively.
Burden	Mothers denied their children were stunted, even asking for measurements on the spot. However, after explaining the program’s benefits, they wanted to be educated. Another burden issue is time mismatch and time-consumption, because it takes 30–60 min for each child’s home visit.
Ethics	Short courses and home visits do not conflict with existing norms, programs, culture, and beliefs, even if mothers and cadres give each other a strengthening of returns to reduce the problem of stunted children.
Perception of effectiveness	This program is beneficial because home visits educate mothers regarding growth monitoring, how to stimulate development, measuring developmental achievements, and appropriate feeding of children. Furthermore, the program support further decreases the prevalence of stunting.
Intervention coherence	Cadres understand the program flow.
Opportunity cost	The home visit program has implications for time, travelling cost, and other resources, along with food ingredients and BKB kits (Indonesian: acronym for Bina Keluarga Balita, or Toddler Family Development) as educational media.
Self-efficacy	Training makes cadres confident; hence, they can solve stunting problems through home visits.

## Data Availability

All data and models of study are available from the corresponding author upon reasonable request.

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
