# Peer review of "Effect of a Short Course on Improving the Cadres’ Knowledge in the Context of Reducing Stunting through Home Visits in Yogyakarta, Indonesia"

_ijerph, 2022, doi:10.3390/ijerph19169843_

Round 1

Reviewer 1 Report

This manuscript needs significantly more information presented.

Abstract:

Comment 1.     Effect size interpretations are needed in the abstract. (Line 28-31)

Introduction:

Comment 1.     The format of reference quotation needs to be adjusted. For example, "[1], [2]"(Line 38), it is suggested to revise it to [1,2].

Comment 2.     Line 47, [[6]]?

Comment 3.     The research is based on ICCM theory design, but I didn't find the corresponding theoretical introduction in the preface. It is suggested to increase the introduction of this theory, and why this theory should be used. Has this theory been used in existing intervention studies? If so, what is the effect?

Comment 4.     Figure 1 is not very clear, so it is suggested to revise it again.

Comment 5.     Abbreviation usage specification: it is suggested to supplement CGM, DG and IYCF. What exactly do they represent? (Line 72,78, etc.) I didn't find the corresponding abbreviation in the introduction, which specifically refers to. You should know that some readers may not understand the Global Strategy for Infant and Young Child Feeding.

Methods:

Comment 1.     The methods require significant more information.

Comment 2.     The process description of the intervention is a bit vague, and a flowchart representation is recommended.

Comment 3.     The meaning of Figure 2 is a little confusing. For example, day-1, Day 2. Do you mean on the first and second day of each week?

Comment 4.     What is the approximate length of each home visit in the intervention? How is supervision and management carried out specifically?

Comment 5.     A detailed presentation of the questionnaire is recommended in the data collection.

Comment 6.     Additionally, you need to include the reliability and validity of the questionnaire in your report. (Line 107-112)

Comment 7.     How were subjects recruited? What are the criteria for subject inclusion? Do they have the right to withdraw from the intervention program?

Results:

Comment 1.     Table 2 shows the results of the t-test. You reported the difference and p-value between the retest and the baseline test. I hope you can indicate the magnitude of the effect (Effect size). You already have the data, so I believe these calculations are very simple for you.

Comment 2.     It would be better to add reporting QIC and OR to the GEE analysis results.

Discussion and Conclusions

Comment 1.     I am having a tough time determining whether the discussion would be fully justified based on the lack of information in the methodology and results as that may dictate how some of the results are presented and interpreted.

Reference:

Comment 1.     It is suggested to increase DOI of journal articles in the revised edition. It is recommended that you carefully check the format of each reference and check the requirements of MDPI references (https://www.mdpi.com/journal/ijerph/instructions). The reference list should include the full title, as recommended by the ACS style guide.

Comment 2.     [9], [27] Missing volume number.

Journal Articles:

1. Author 1, A.B.; Author 2, C.D. Title of the article. Abbreviated Journal Name Year, Volume, page range.

Comment 3.     [35] Typing mistake

Author Response

Dear reviewers,

We appreciate for reviewers’ comments and suggestions. In this document, we provide our responses to the reviewers' valuable comments and suggestions. We would be glad if you could have our manuscript reviewed again and provide us with comments. Thank you

Reviewer 2 Report

Thank you that you give me the opportunity to review this manuscript Effect of Short Course on Improving the Cadres’ Knowledge in the Stunting Reducing Context through Home Visits in Yogygkarta, Indonesia”.

The authors presented the interesting, still actual topic of stunting among Indonesian children.

The manuscript  is noteworthy, and worth further research. The appropriate tables and figures have been provided. The article is easy to read and logically structured.  The methods are adequately described. The authors used appropriate statistical methods. The conclusions are consistent with the presented evidence and arguments.

Some of the comments on this manuscript.

  1. Abstract

Line 19: Stunting is primarily a public health concern in LMIC. The authors use an abbreviation that is not explained anywhere. Please correct it.

Line 28: On post-tests 1 and 2, cadres' knowledge of IYCF… the same, Please correct it.

Key words: The authors use an abbreviation IYCF, but better will be add “ stunting”

  1. Introduction

Line 40: Furthermore, after Bali and Jakarta at 17.3% and 10.9%, respectively, Yogyakarta is the province with the 3rd lowest prevalence of stunting at 17.3% [4] .  Maybe you should add in Indonesia. Furthermore, after Bali and Jakarta at 17.3% and 10.9%, respectively, Yogyakarta 40

is the province with the 3rd lowest prevalence of stunting at 17.3% in Indonesia [4].

The authors write for the international journal. And for example Bali (they think about an island belonging to Indonesia), but this sentence does not explain it. The preceding sentence or the following sentence does not mention Indonesia too.

Bali also is, for example, a beautiful town in Crete, an island in the Mediterranean Sea that a reader comes from Europe may think of.

Line 38 and 40: [1], [2].  Please change to [1,2]  or [2], [3]  change to  [2,3].

Line 47: [[6]]. Double parenthesis , please correct

Line 72:  … CGM, DGM and IYCF..   Please explain all these abbreviations.

 3.Material and Methods

All abbreviations (CGM, CDM, IYCF), that includes in Figure 2 should be explained below the table. 

4.Discussion

At the beginning of the discussion, the authors describe the Cadres. But in line 46: Cadres are community health workers selected by the residents based on their  ability, integrity, loyalty, and commitment to improving community health status [[6]].

It follows that these people do not have medical education, only completed training.

But in the introduction in line 46 cadres the authors call health workers, which in middle and highly developed countries means people with medical education.

In order not to mislead the reader, it is necessary to clarify the Cadres.

Line 282: …are taught [42], [43], change to  [42,43].

5. References

Please check the literature.

e.g., 19. Permendes RI. Penetapan Prioritas Penggunaan Dana Desa Tahun 2018.2017, is it sure it is correctly given.

Author Response

Dear reviewers,

Enclosed is the revised manuscript that we are submitting for reconsideration for publication in International Journal of Environmental Research and Public Health. We have revised our manuscript in accordance with the reviewers’ comments and suggestions. In this document, we provide our responses to the reviewers' valuable comments and suggestions. We would be glad if you could have our manuscript reviewed again and provide us with comments

Round 2

Reviewer 1 Report

Thanks for the correction. It has clarified some of my questions about this study. The revision is better than the original.However, there are still some small problems that need to be improved.

(1)Figures 1 and 2 are still unclear and I suggest replacing them with clear ones.

(2)The web citation in the reference suggests adding the date of acquisition.

Author Response

Dear Author, 

Enclosed is the revised manuscript that we are submitting for reconsideration for publication in International Journal of Environmental Research and Public Health. We have  revised our manuscript in accordance with the reviewers’ comments and suggestions. 

The revision are : 

Figures 1

Figure 2

Date of acquisition of web citation in the reference 

Thank you

Best Regards
